



# LES study on turbulent dust deposition and its dependence on atmospheric boundary-layer stability

Xin Yin[1], Cong Jiang[1], Yaping Shao[1], Ning Huang[2], Jie Zhang[2]

[1] Institute for Geophysics and Meteorology, University of Cologne, Cologne, Germany
[2] Key Laboratory of Mechanics on Disaster and Environment in Western China, Lanzhou University, Lanzhou, China

*Correspondence to*: Jie Zhang (zhang-j@lzu.edu.cn)

**Abstract.** It is increasingly recognized that atmospheric boundary-layer stability (ABLS) plays an important role in aeolian processes. While the effects of ABLS on dust emission have been documented in several
studies, those on dust deposition are less well studied. By means of large-eddy simulation, we investigate how ABLS influences the probability distribution of surface shear stress and hence dust deposition. Statistical analysis of the model results reveals that the shear stress can be well approximated by using a Weibull distribution and the ABLS influences on dust deposition can be estimated by considering the shear stress fluctuations. The model-simulated dust depositions are compared with the predictions of a dust-deposition
scheme and measurements, and the findings are then used to improve the dust-deposition scheme. This research represents a further step towards developing dust schemes that account for the stochastic nature of dust processes.

**Keywords:** Dust deposition, Atmospheric boundary-layer stability, Surface shear stress, Weibull distribution, Stochastic dust process

## 1 Introduction

Dry deposition is the removal of particulates and gases at the air-surface interface by turbulent transfer and gravitational settling (Sehmel, 1980; Droppo, 2006; Hicks et al., 2016). Because it is the only process for the removal of particles from the atmosphere in the absence of precipitation, developing reliable methods for estimating dry deposition of particles has attracted much interest since the early 1940s (Gregory, 1945;
Chamberlain, 1953; Slinn and Slinn, 1980; Slinn, 1982; Zhang et al., 2001; Petroff and Zhang, 2010; Zhang and Shao, 2014; Seinfeld et al., 2016). Several particle-deposition schemes have been proposed (e.g., Slinn, 1982; Zhang and Shao, 2014) for regional/global models, which are driven by using several environmental parameters, including the Reynolds surface shear stress (typically averaged over 15-30 min). However, field observations indicate that the use of Reynolds stress as the only wind-related parameter in such schemes may
not be sufficient to achieve accurate estimates of particle deposition, because of the nonlinear relationship between deposition velocity and wind shear. The observations using the eddy correlation method show that particle-deposition velocity has strong spatiotemporal variations associated with the fluctuations of wind speed (Connan et al., 2018; Damay et al., 2009; Lamaud et al., 1994; Wesely et al., 1983, 1985). It is also



observed that when the background wind speeds are similar, deposition velocity under convective conditions
is in general larger than under neutral and stable conditions. Pellerin et al. (2017) suggested that cospectral similarities exist between heat and particle-deposition fluxes and that atmospheric turbulence plays a role in dust deposition. It is therefore necessary to find a link between instantaneous wind and particle deposition and to correctly represent this link in particle-deposition schemes, i.e., to introduce and account for the effect of turbulence on particle deposition.

Models for turbulent dust emission ( Klose and Shao, 2012, 2013) and sand saltation (Liu et al., 2018; Li et al., 2020; Rana et al., 2020) have been developed, but to our knowledge, although turbulent dust deposition is now perceived to be important, a scheme is yet to be constructed for its quantitative estimate.

The turbulent wind flow in a particle-deposition scheme is reflected in the turbulent shear stress (or vertical momentum flux). It is well-known that apart from gravitational settling, particle deposition is driven by
turbulent diffusion which is intimately related to the vertical momentum transfer in the atmospheric boundary layer (ABL) (Wyngaard, 2010). Based on the Prandtl mixing-length theory, the shear stress can be parameterized in neutral conditions. However, it is known that for a given mean wind speed (at a reference height) in the ABL, both the mean value and the perturbations of shear stress depend on the atmospheric boundary-layer stability (ABLS), for instance, shear stress shows generally larger fluctuations in convective
ABLS. Klose and Shao (2012) pointed out that, under convective conditions, large eddies have coherent structures of dimensions comparable to boundary-layer depth, which are efficient entities in generating localized momentum fluxes to the surface. Although the eddies only occupy fractions of time and space, the momentum fluxes to these fractions can be many times the average. Hicks et al. (2016) mentioned that ABLS is of immediate concern in the micrometeorological community, because of its influences on the
intermittency, gustiness and diurnal cycle of particle deposition. Similar to dust emission and sand saltation, intermittent dust deposition also occurs as a result of fluctuating surface shear stress. The current particle-deposition schemes only consider the mean behavior of wind, and how this mean behavior varies with ABLS via the Monin-Obukhov similarity theory, (Monin et al., 2007; Monin and Obukhov, 1954), but not the fluctuations of the associated shear stress and how they vary with ABLS.

We argue that focusing only on the effects of ABLS on mean wind is insufficient to model particle deposition accurately. In this study, we explore the influences of ABLS on the turbulent behavior of particle deposition and attempt to improve an existing particle deposition scheme. A large-eddy simulation (LES) model is used to simulate turbulence and particle deposition under various ABLS conditions. The dust particle depositions simulated using the LES model and predicted using the particle-deposition scheme of Zhang and Shao (2014,
ZS14 hereafter) are compared with each other and with measurements. Here, we address the following three issues: (1) How ABLS affects the probability distribution of surface shear stress; (2) How ABLS impacts on particle deposition; and (3) How the ZS14 scheme can be improved to account for the ABLS effect. On this basis, an improvement to the ZS14 scheme (also applicable to other schemes) is proposed. The remaining part of the paper is organized as follows: Sect. 2 gives a brief description of the Weather Research and





Forecast – Large-Eddy Simulation Model with Dust module (WRF-LES/D), the ZS14 scheme, and the design of the numerical experiments. Sect. 3 discusses the findings of the numerical simulations and the improvement to the ZS14 scheme. The concluding remarks are given in Sect. 4.

## 2. Model/Method

### 2.1 WRF-LES/D

The WRF-LES/D used here is initially developed by Shao et al. (2013) and Klose and Shao (2013) by coupling the WRF-LES (Moeng et al., 2007; Skamarock et al., 2008) with a land-surface module and dust module. As demonstrated in the earlier studies, WRF-LES/D is a reasonably well-established system for applications to simulating turbulence, turbulent dust emission and transport for various ABLS conditions. WRF-LES is a three-dimensional and non-hydrostatic model for fully compressible flow. The model

separates the turbulent flow into a grid-resolved component and a subgrid component. The *k-l* subgrid closure (Deardorff, 1980) together with the TKE equation (Skamarock et al., 2008) based on nonlinear backscatter and anisotropic (Kosović, 1997; Mirocha et al., 2010) are used here. The governing equations in WRF-LES/D include the equations of motion, continuity equation, enthalpy equation, equation of state and the dust conservation equation, as shown below

$$\frac{\partial u_i}{\partial t} + \frac{\partial u_i u_j}{\partial x_j} = -\delta_{i3} g + \varepsilon_{ij3} f u_j - \frac{1}{\rho_a}\frac{\partial p}{\partial x_i} + v\frac{\partial^2 u_i}{\partial x_j x_j} - \frac{1}{\rho_a}\frac{\partial \tau_{ij}}{\partial x_j} \tag{1}$$

$$\frac{\partial \rho_a}{\partial t} + \frac{\partial \left(\rho_a u_j\right)}{\partial x_j} = 0 \tag{2}$$

$$\frac{\partial c_p T}{\partial t} + \frac{\partial \left(c_p u_j T\right)}{\partial x_j} = \frac{\partial H_j}{\partial x_j} + s_T \tag{3}$$

$$p = \rho_a R_a T \tag{4}$$

$$\frac{\partial c}{\partial t} + u_j \frac{\partial c}{\partial x_j} - w_t \frac{\partial c}{\partial z} = -\frac{\partial F_j}{\partial x_j} + s_r \tag{5}$$

where $u_i$ $(u, v, w)$ is grid-resolved flow velocity along $x_i$ $(x, y, z)$ refer to the streamwise, spanwise, and vertical directions, respectively; $g$ is the acceleration due to gravity; $\rho_a$ is the air density; $f$ is the Coriolis parameter; $p$ is pressure; $\tau_{ij}$ is subgrid stress tensor modeled using an eddy viscosity approach where the eddy viscosity is represented as the product of a length scale and a velocity scale characterizing the subgrid-scale (SGS) turbulent eddies, with the velocity scale being derived from the SGS TKE and the length scale

from the grid spacing; $v$ is the kinematic viscosity; $\delta_{ij}$ is the Kronecker operator and $\varepsilon_{ij}$ is the alternating operator; $c_p$ is the specific heat of air at constant pressure; $T$ is air temperature; $H_j$ is the $j$th component of subgrid heat flux; $c$ is dust concentration; $w_t$ is dust particle terminal velocity; $F_j$ is the $j$th component of subgrid dust flux; $s_T$ and $s_r$ are the source or sink terms for heat and particles, respectively. The eddy



diffusivity is obtained using eddy viscosity dividing the Prandtl number. For the surface layer, an important

parameterization to solve the governing equations for high-Reynolds-number turbulence is embedded in the surface boundary condition, which computes the instantaneous local surface shear stress using the bulk transfer method (Kalitzin et al., 2008; Kawai and Larsson, 2012; Piomelli et al., 2002; Zheng et al., 2020) as follows,

$$\tau = \rho_a K_m \frac{\partial V}{\partial z} \qquad (6)$$

with

$$K_m = \frac{k u_* z}{\varphi_m} \qquad (7)$$

where $K_m$ being eddy viscosity and $\varphi_m$ being the MOST stability function, $V = \sqrt{u^2 + v^2}$. Even though Shao et al. (2013) questioned the application of the MOST in LES, it is still used here, as our emphasis is on the variance of shear stress in the simulation domain. Several land-surface models (LSMs) can be selected

(e.g., Chen and Dudhia, 2000; Pleim and Xiu, 2003) in WRF-LES/D, and the 5-layer thermal diffusion (Dudhia, 1996) is used in this study. Surface heat flux in this study is artificially given. In addition, we denote surface heat flux as $H_0$ and dust dry deposition flux on grand in each grid as $F_d$.

**2.2 Particle-deposition scheme of ZS14**

The dust particle deposition on the surface is more complicated than momentum flux as the dust concentration

changing close to the surface is unclear. To solve the dust conservation equation, Eq. (5), the emission and deposition fluxes at the surface need to be specified. The problem of dust emission has been dealt with elsewhere (Shao, 2004; Klose and Shao, 2013) and is not considered here. For our purpose, dust emission is assumed to be zero. This section gives the parameterization scheme of surface settlement proposed by ZS14. The detail of the scheme is as described in ZS14, only the main results are given here for completeness. In

general, we can express dust deposition flux $F_d$ as

$$F_d = -\left(K_p + k_p\right)\frac{\partial c}{\partial z} - w_t \cdot c \qquad (8)$$

where $K_p$ and $k_p$ are eddy diffusivity and molecular diffusivity, respectively. By analogy with the bulk-transfer formulation of scalar fluxes in ABL, $F_d$ can be parameterized as

$$F_d = -V_d(z) \cdot c(z) \qquad (9)$$

where $c(z)$ is the dust concentration at height $z$ (the center height of the lowest model level in this study), $V_d(z)$ is the corresponding dry deposition velocity.

The surface layer is divided into an inertial layer and a roughness layer. Integrating Eq. (8) in inertial layer and substitute Eq. (9) into it, $V_d(z)$ is obtained:





$$V_d(z) = \left( r_g + \frac{r_s - r_g}{\exp(r_a / r_g)} \right)^{-1}$$
(10)

with $r_a$ being the aerodynamic resistance for the inertial layer. Using the MOST, we have

$$r_a = \frac{S_{cT}}{k u_*} \left[ \ln\left( \frac{z - z_d}{h - z_d} \right) - \psi_m \right]$$
(11)

where $z_d$ is the displacement height, $h$ is the height of roughness element $\psi_m$ is the integral of stability function in the inertial layer, $S_{cT} = K_m / K_p$ (Csanady, 1963), and $\kappa$ is the von Karman constant. The gravitational resistance $r_g$ is defined as the reciprocal of the gravitational settling and depends mainly on particle size and

density. A free-falling particle is subject to gravitational and aerodynamic drag forces. When these forces are in equilibrium, the gravitational settling velocity reaches the terminal velocity given by the Stokes formula

$$w_t = \frac{C_u \rho_p D_p^2 g}{18 \mu_a} = r_g^{-1}$$
(12)

where $D_p$ is the particle diameter, $\rho_p$ is the particle density, $\mu_a$ is the air dynamic viscosity, $C_u$ is the Cunningham correction factor that accounts for the slipping effect affecting the fine particles.

In the roughness layer, the collection process is reflected in collection resistance, defined by $r_s = -\frac{c(h)}{F_d}$ with an assumption of dust concentration is zero on roughness elements or ground. In addition to the meteorological factors and land-use category, Zhang and Shao (2014) established a relationship between aerodynamic and surface-collection processes by using an analogy between drag partition and deposition flux partition, which can describe surface heterogeneity.

$$r_s^{-1} = R \cdot \frac{\tau}{\rho_a u_h} \left( \frac{E}{C_d} \frac{\tau_c}{\tau} + \left( 1 + \frac{\tau_c}{\tau} \right) S_c^{-1} + 10^{-\frac{3}{\hat{T}}} \right) + w_t$$
(13)

Here, $R$ is the reduction of collection caused by particle rebound, $E$ is the collection coefficient of the roughness elements, which includes the contributions of Brownian motion, impaction and interception, $S_c$ is the Schmidt number which is the ratio of air viscosity to molecular diffusion, $u_h$ is the wind speed at the top of roughness layer, $C_d$ is the drag coefficient for isolated roughness element, $10^{-\frac{3}{\hat{T}}}$ represents the turbulent

impaction efficiency with $\hat{T}$ being the dimensionless particle relaxation time. The ratio $\tau_c / \tau$ describes the drag partition with $\tau_c$ being the pressure drag (the force exerted on roughness elements) and can be calculated according to Yang and Shao (2006) as





$$\frac{\tau_c}{\tau} = \frac{\beta\lambda_e}{1+\beta\lambda_e} \tag{14}$$

and

$$\lambda_e = \frac{\lambda}{(1-\eta)^6} \cdot \exp\left(-\frac{\lambda}{10\cdot(1-\eta)^6}\right) \tag{15}$$

with $\beta$ (= 200) is the ratio of the pressure drag coefficient for isolated roughness element to that of bare surface, $\lambda$ is the frontal area index of the roughness elements, $\eta$ is the basal area index or the fraction of cover. From Eqs. (10)-(15), it can be seen that $V_d$ and $\tau$ are nonlinearly related. As example, for the particle of diameter 1 μm, analysis shows that when $\tau$ is small, $V_d$ is dominated by $w_t$. As $\tau$ increases, $w_t$ and $\tau$ are both

important in $V_d$. As $\tau$ increases further, the effect of $\tau$ becomes much larger than gravity settling, thus the $V_d$ is mainly determined by $\tau$.

**2.3 Simulation Set-up**

Numerical experiments are carried out with WRF-LES/D for various atmospheric stability and background-wind conditions for two different roughness lengths (Table 1). The domain of the simulation is

$2000 \times 2000 \times 1500$ m³ and the number of grid points is $200 \times 200 \times 90$ corresponding to a horizontal resolution $\Delta x = \Delta y = 10$ m. The Arakawa-C staggered grid is used. The depth of the lowest model layer is 1 m and the grid above is stretched following a logarithmic function of $z$. The simulation time is 90 minutes with a time step of 0.05 s and the output interval is 10 s. The first 30 minutes of the simulation is the model spin-up time and the data of the remaining 60 minutes are used for the analysis.

For model initialization, the wind and dust concentration (Chamberlain, 1967; Monin, 1970; Kind, 1992) are assumed to be logarithmic in the vertical and uniform in the horizontal direction. For each experiment, a constant surface heat flux is specified. A 300 m deep Rayleigh damping layer is used at the upper boundary with a damping coefficient of 0.01. The wind speed at the top boundary, $U$, is given in Table 1. The surface heat flux, $H_0$, increases from -50 to 600 W m⁻², and for each surface heat flux, the wind conditions increase

from 4 to 16 m s⁻¹ in Exp (1-20) and from 5.44 to 18.12 m s⁻¹ in Exp (21-35). The roughness length $z_0$ for sand surface used in Exp (1-20) is 0.153 mm following wind tunnel experiment (Zhang and Shao, 2014) but 0.76 mm in Exp (21-35) according to field observation (Bergametti et al., 2018). The lateral boundary conditions are periodic, which allows the simulation of a well-developed boundary layer. The vertical scaling velocity is estimated using heat flux, $w_* = \left(\dfrac{g}{\bar{\theta}}\dfrac{H_0}{\rho_a c_p}z_l\right)^{1/3}$ , with $\bar{\theta}$ being the mean potential temperature

and $z_l = 1000$ m is the boundary layer inversion height. Usually, $w_*$ is not used for stable ABLS, but used here as an indicator for the suppression of turbulence by negative buoyancy.



Table 1: List of numerical experiments with $z_0 = 0.153$ mm for Exp (1-20) in wind tunnel experiments (Zhang and Shao, 2014) and $z_0 = 0.76$ mm for Exp (21-35) in field observation (Bergametti et al., 2018) for sand surface.

| $z_0 = 0.153$ mm | | $z_0 = 0.76$ mm | | | |
| NAME | $U$ (m s$^{-1}$) | NAME | $U$ (m s$^{-1}$) | $H_0$ (W m$^{-2}$) | $w_*$ (m s$^{-1}$) |
|---|---|---|---|---|---|
| EXP1 | 4 | EXP21 | 5.44 | -50 | -1.12 |
| EXP2 | 8 | EXP22 | 10.87 | -50 | -1.12 |
| EXP3 | 12 | EXP23 | 18.12 | -50 | -1.12 |
| EXP4 | 16 | -- | -- | -50 | -1.12 |
| EXP5 | 4 | EXP24 | 5.44 | 0 | 0 |
| EXP6 | 8 | EXP25 | 10.87 | 0 | 0 |
| EXP7 | 12 | EXP26 | 18.12 | 0 | 0 |
| EXP8 | 16 | -- | | 0 | 0 |
| EXP9 | 4 | EXP27 | 5.44 | 200 | 1.77 |
| EXP10 | 8 | EXP28 | 10.87 | 200 | 1.77 |
| EXP11 | 12 | EXP29 | 18.12 | 200 | 1.77 |
| EXP12 | 16 | -- | -- | 200 | 1.77 |
| EXP13 | 4 | EXP30 | 5.44 | 400 | 2.23 |
| EXP14 | 8 | EXP31 | 10.87 | 400 | 2.23 |
| EXP15 | 12 | EXP32 | 18.12 | 400 | 2.23 |
| EXP16 | 16 | -- | -- | 400 | 2.23 |
| EXP17 | 4 | EXP33 | 5.44 | 600 | 2.55 |
| EXP18 | 8 | EXP34 | 10.87 | 600 | 2.55 |
| EXP19 | 12 | EXP35 | 18.12 | 600 | 2.55 |
| EXP20 | 16 | -- | -- | 600 | 2.55 |

## 3. Results

### 3.1 Turbulent shear stress

In the first set of the analysis, we examine the impact of atmospheric stability on shear stress fluctuations. Early dust deposition studies considered only the time average of surface shear stress, $\tau_r$, with the assumption that shear stress is horizontally homogeneous. In WRF-LES/D, the corresponding mean resultant shear stress $\tau_r$ can be obtained as below:

$$\tau_r = \sqrt{\overline{\tau}_{xz}^2 + \overline{\tau}_{yz}^2} \tag{16}$$

The shorthand notation $\overline{f} = \dfrac{1}{N_x N_y N_t} \sum_{n_x n_y n_t} f(n_x, n_y, n_t)$ is introduced to represent the space and time

average over the simulation domain and time period (hereafter ensemble mean) with $N_x$ (=200) and $N_y$ (=200) are the numbers of grid points in the $x$- and $y$-direction, respectively, and $N_t$ (=360) the time steps of model output.

Figure 1a-c show the instantaneous shear stress, $\tau$, of a sample grid ($n_x = 198$, $n_y = 41$) over a one-hour period for the runs with $z_0 = 0.153$ mm, $U = 4$ m s$^{-1}$ and various ABL stabilities ($H_0 = 0, 200, 600$ W m$^{-2}$). Figure 1d-f is same as Fig. 1a-c, but for $U = 16$ m s$^{-1}$. The panel shows that $\tau$ is not a constant, and the mean resultant shear stress, as well as the shear stress fluctuations, increase with increasing atmospheric instability. In



addition, the insert plots in Fig. 1 show that the autocorrelation function, ACF, is oscillated during decrease.
The oscillation periodicity is longer under weak wind conditions (Fig. 1a-c) than strong wind (Fig. 1d-f). The ACF in neutral conditions decreases rapidly than in convective conditions. Recall the definition of coherent motion given by Robinson (1991) - the correlation of variables over a range of long time larger than the smallest scales of flow is an evidence of coherent oscillating motion. Thus, the regular oscillation and a long time correlation of $\tau$ are closely related to the evolvement of the coherent structure. This indicates that in a
convective ABL, stronger large-scale coherent structures exist even under weak wind conditions.

To gain insight into the behavior of the unsteady shear stress field, we introduce the turbulence intensity of surface shear stress (TI-S) defined as the ratio of the standard deviation of fluctuating surface shear stress, $\sigma_\tau$, to the mean resultant stress $\tau_r$, i.e., $\sigma_\tau/\tau_r$. Analysis shows that $\sigma_\tau/\tau_r$ increases as atmospheric conditions become more unstable and decreases with wind speed (e.g., Fig. 1). High wind speeds tend to force the ratio
to be more similar to that in neutral ABLs, as the mean-wind induced shear stress becomes dominant over the large-eddy induced shear-stress fluctuations. For a weak TI-S, $\tau$ is dominated by $\tau_r$ and the stress fluctuations are small compared to $\tau_r$. As TI-S increases, the contribution of momentum transport by large eddies becomes significant because in unstable ALBS, buoyance generated large eddies penetrate to high levels and intermittently enhance the momentum transfer to the surface.

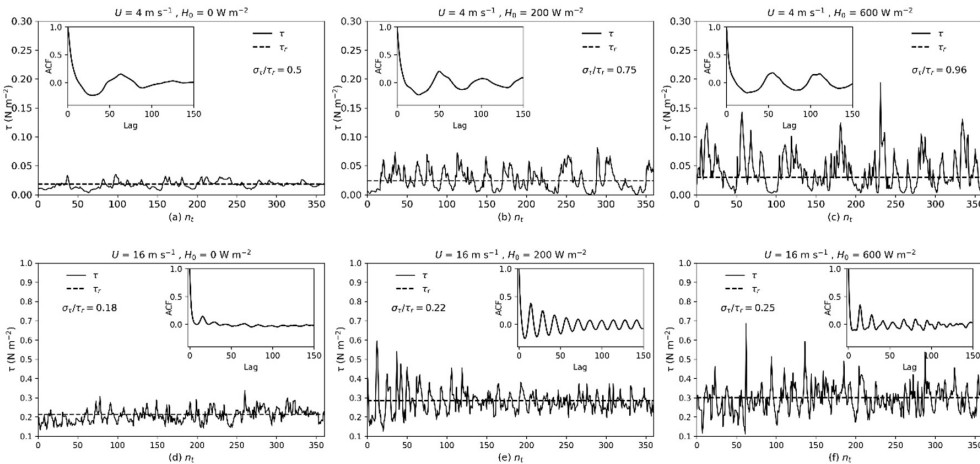

**Figure 1.** Time evolutions of surface shear stress $\tau$ with different $H_0$ values and $z_0 = 0.153$ mm at the grid point $n_x = 198$ and $n_y = 41$ **(a-c)** for $U = 4$ m s$^{-1}$; **(d-f)** for $U = 16$ m s$^{-1}$; the insert plots are the autocorrelation functions of $\tau$.

The intermittent surface shear stress can directly cause localized dust deposition. Therefore, dust deposition is also intermittent in space and time. However, to our knowledge, in existing dust-deposition schemes (e.g.,
ZS14 used here), the dust-deposition velocity is calculated using only the mean resultant shear stress $\tau_r$ instead of the instantaneous shear stress. We denote this deposition velocity as $V_{d,\tau_r}$. The mean deposition

velocity simulated by WRF-LES/D, denoted as $V_{d,LES}$, is estimated via the ratio of the ensemble mean of dust deposition flux and the ensemble mean of dust concentration:

$$V_{d,LES} = -\frac{\bar{F}_d}{\bar{c}}$$

(17)

which is consistent with the methods commonly used in field observations and wind-tunnel experiments.

Figures 2a and 2b, with the same wind conditions as for Fig.1a-c and 1d-f, show the time evolution of the instantaneous deposition velocity $V_d$ for particles of diameter 1.46 μm and surface heat flux $H_0 = 600$ W m$^{-2}$. As shown, the fluctuation behavior of $V_d$ is consistent with that of $\tau$. Moreover, Fig. 2a shows a substantial difference between $V_{d,LES}$ and $V_{d,\tau_r}$, while Fig. 2b shows $V_{d,\tau_r}$ is similar with $V_{d,LES}$. This suggests that the

ZS14 scheme can more accurately estimate the deposition velocity for weak TI-S but underestimates the deposition velocity for strong TI-S. The reason for this is that in the case of strong TI-S, dust deposition caused by the gusty wind plays an important role as $V_d$ and $\tau$ are non-linearly related, which is not reflected in $V_{d,\tau_r}$. Since $\tau$ fluctuates and sometimes strongly, a bias always exists in conventional dust-deposition schemes and the magnitude of the bias depends on turbulence intensity. Therefore, in order to estimate dust

deposition accurately, we need to first describe and parameterize the shear stress.

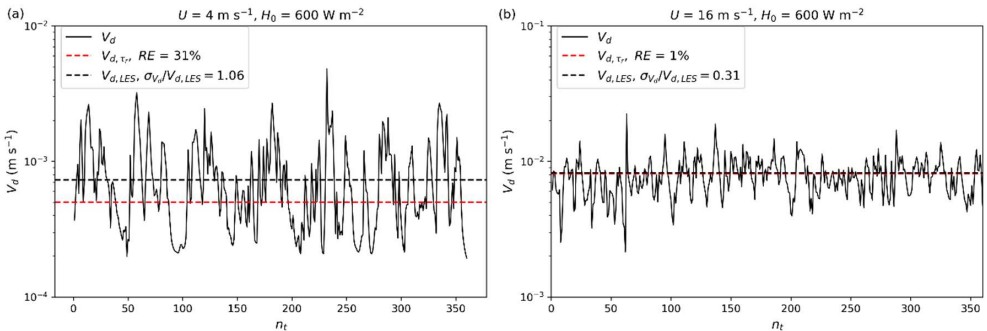

**Figure 2.** Time evolutions of deposition velocity $V_d$ at grid point $n_x = 198$, $n_y = 41$ when $H_0 = 600$ W m$^{-2}$, $z_0$=0.153 mm and **(a)** $U = 4$ m s$^{-1}$ and **(b)** $U = 16$ m s$^{-1}$. $RE = \left|\frac{V_{d,LES}-V_{d,\tau r}}{V_{d,LES}}\right| \times 100\%$ is the relative error between $V_{d,\tau_r}$ and $V_{d,LES}$, $\sigma_{V_d}/V_{d,LES}$ is the ratio of the standard deviation of simulated instantaneous deposition velocity $V_d$ and mean deposition

velocity, $V_{d,LES}$.

As a main predisposing factor for aeolian processes, turbulent shear stress has attracted much attention (e.g., Klose et al., 2014; Li et al., 2020a; Liu et al., 2018; Rana et al., 2020; Zheng et al., 2020). Similar to previous studies, we use the probability density function $p(\tau)$ to characterize the stochastic variable $\tau$. Figure 3 shows that the variability of $\tau$ increases as atmospheric instability increases. The statistic moments of $\tau$, including

its mean resultant value $\tau_r$, standard deviation $\sigma_\tau$, skewness $\gamma_l$ of Exp (1-20) are listed in Table 2. $\sigma_\tau$ and $\tau_r$ increases with increased instability, and the distribution is positively skewed. Positive skewness is characterized by the distribution having a longer positive tail as compared with the negative tail and the





distribution appears as a left-leaning (i.e., tends toward low values) curve. This indicates that large negative fluctuations are not as frequent as large positive fluctuations. The data also shows $\gamma_1$ generally shows a

downward trend as TI-S decreases, which is consistent with (Monahan, 2006), i.e., as TI-S decreases, $p(\tau)$ becomes increasingly Gaussian.

**Table 2.** Statistics of shear stress for numerical experiments Exp (1-20).

| NAME | $H_0$ | $U$ | $\tau_r$ | $\sigma_\tau$ | $\sigma_\tau/\tau_r$ | $\gamma_1$ | $\alpha$ | $\beta$ |
|------|------|-----|----------|----------|-----------|-------|-------|-------|
| EXP1 | -50 | 4 | 0.0156 | 0.0086 | 0.554 | 1.902 | 2.026 | 0.011 |
| EXP2 | | 8 | 0.0295 | 0.0096 | 0.327 | 1.573 | 3.154 | 0.023 |
| EXP3 | | 12 | 0.0524 | 0.0115 | 0.22 | 1.029 | 3.923 | 0.044 |
| EXP4 | | 16 | 0.1009 | 0.0158 | 0.157 | 0.835 | 4.819 | 0.09 |
| EXP5 | 0 | 4 | 0.0185 | 0.0093 | 0.5 | 1.896 | 3.049 | 0.017 |
| EXP6 | | 8 | 0.0604 | 0.0151 | 0.25 | 1.142 | 5.004 | 0.055 |
| EXP7 | | 12 | 0.1315 | 0.0266 | 0.202 | 0.166 | 5.383 | 0.122 |
| EXP8 | | 16 | 0.2136 | 0.038 | 0.178 | 0.087 | 6.191 | 0.196 |
| EXP9 | 200 | 4 | 0.024 | 0.018 | 0.75 | 1.142 | 1.56 | 0.025 |
| EXP10 | | 8 | 0.0812 | 0.0325 | 0.4 | 1.02 | 3.022 | 0.076 |
| EXP11 | | 12 | 0.1676 | 0.0451 | 0.269 | 0.512 | 4.078 | 0.156 |
| EXP12 | | 16 | 0.2848 | 0.0624 | 0.219 | 0.766 | 5.214 | 0.259 |
| EXP13 | 400 | 4 | 0.026 | 0.0248 | 0.955 | 1.127 | 1.302 | 0.03 |
| EXP14 | | 8 | 0.0825 | 0.0372 | 0.451 | 0.646 | 2.513 | 0.081 |
| EXP15 | | 12 | 0.1728 | 0.0522 | 0.302 | 0.677 | 3.776 | 0.160 |
| EXP16 | | 16 | 0.2992 | 0.0646 | 0.216 | 0.289 | 5.214 | 0.278 |
| EXP17 | 600 | 4 | 0.0299 | 0.0287 | 0.96 | 1.083 | 1.303 | 0.035 |
| EXP18 | | 8 | 0.0894 | 0.0424 | 0.474 | 0.715 | 2.472 | 0.089 |
| EXP19 | | 12 | 0.1767 | 0.0604 | 0.342 | 0.614 | 3.252 | 0.167 |
| EXP20 | | 16 | 0.3003 | 0.0739 | 0.246 | 0.511 | 4.493 | 0.277 |



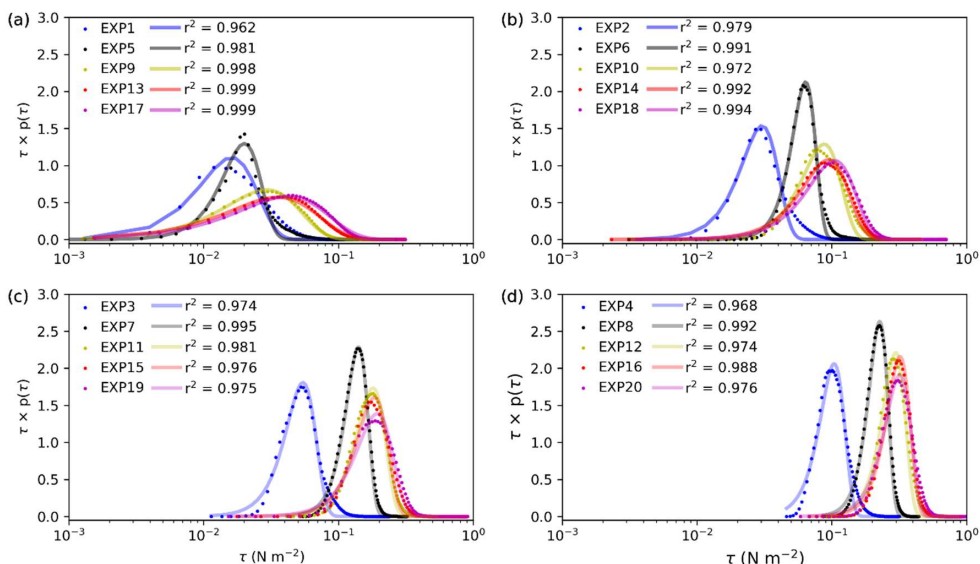

**Figure 3.** Probability density functions derived from WRF-LES/D simulated surface shear stress (dots) and the corresponding fitted Weibull density functions (solid lines, $r^2$ is the coefficients of determination) for different surface heat fluxes and different wind speeds: **(a)** $U = 4$ m s$^{-1}$, **(b)** $U = 8$ m s$^{-1}$, **(c)** $U = 12$ m s$^{-1}$, **(d)** $U = 16$ m s$^{-1}$ with $z_0 = 0.153$ mm.

The parameterization of surface shear stress has attracted intense interests, as example, Klose et al. (2014) reported that $\tau$ in unstable conditions is Weibull distributed based on large-eddy simulations. Shao et al. (2020) found that $p(\tau)$ is skewed to small $\tau$ values (i.e., positively skewed) based on field observations. Li et al. (2020) suggested that $\tau$ in neutral conditions is Gauss distributed based on a wind-tunnel experiment. Colella and Keith (2003) explained that in turbulent shear flows, the non-linear interaction between the eddies gives rise

to a departure from Gaussian behavior. Our results show that the Gaussian approximation is inadequate in representing the skewed $p(\tau)$, especially for the conditions of strong turbulence intensity (e.g., unstable cases in Fig. 3a). Therefore, $p(\tau)$ here is approximated using a Weibull distribution, i.e.,

$$p(\tau) = \frac{\alpha}{\beta}\left(\frac{\tau}{\beta}\right)^{\alpha-1} \exp\left(-(\tau/\beta)^{\alpha}\right) \qquad (18)$$

where $\alpha$ and $\beta$ are the shape and scale parameters, respectively. The values of $\alpha$ and $\beta$ for the numerical

experiments Exp (1-20) are listed in Table 2. It can be seen that both $\alpha$ and $\beta$ depend on wind speed and atmospheric stability. However, $\beta$ is mainly determined by wind conditions when wind is strong, while it is affected by ABL stability when wind is weak. The behavior of $\alpha$ and $\beta$ are shown in Fig. 4. In both stable and unstable atmospheric conditions, analysis shows that the scale parameter $\alpha$ is related to ABL stability as the power of $|1/L_o|$ where $L_o$ is the Monin-Obukhov length. Fig. 4a shows that $\alpha$ decreases with the $|1/L_o|$,

satisfying approximately Eq. (19). For neutral conditions, $L_o$ goes to infinity, Eq. (19) no longer applies. Therefore, the shape parameter obtained by the fitting was directly used for pdf reproduction for the neutral



cases instead of the approximated $\alpha$ used for stable and unstable conditions. As Fig. 4b shows, the $\beta$ parameter increases almost linearly with $u_{*r}^2 + 0.001 \cdot w_*^2$ but can be best approximated using Eq. (20) with $u_{*r} = \sqrt{\tau_r/\rho_a}$.

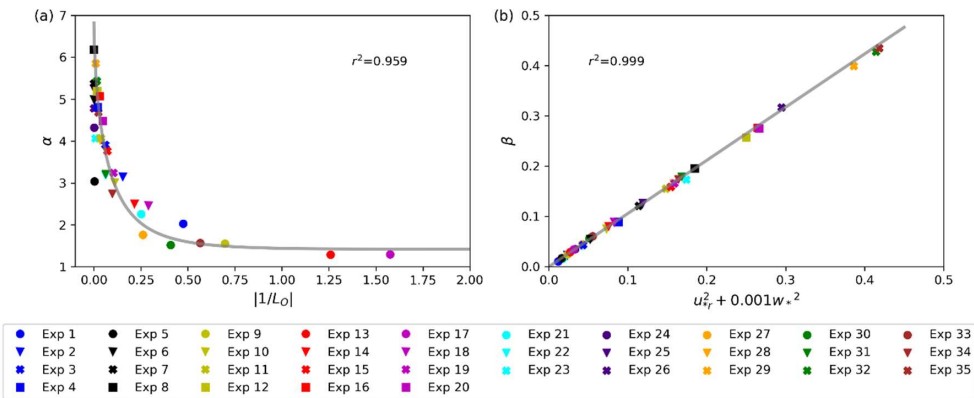

**Figure 4. (a)** Dependency of the shape parameter $\alpha$ on $L_o^{-1}$ for all numerical experiments Exp (1-35); **(b)** Dependency of scaling parameter $\beta$ on $\left(u_{*_r}^2 + 0.001 w_*^2\right)$ for Exp (1-35).

$$\alpha = 5.39 \cdot \exp\left(-5.43\left(\frac{1}{L_o}\right)^{2/3}\right) + 1.42 \tag{19}$$

$$\beta = 1.058 \cdot (u_{*_r}^2 + 0.001 w_*^2) \tag{20}$$

Using Eqs. (18)-(20), we can approximately describe the turbulent surface shear stress in non-neutral cases.

### 3.2 Improvement to dust deposition scheme

Figure 5a shows the performances of WRF-LES/D by comparing the simulated deposition velocity, $V_{d,LES}$, with wind tunnel experiments (Zhang and Shao, 2014) and field observation (Bergametti et al., 2018). The observed data are measured under neutral conditions and similar wind flow. As shown, the simulation results agree well with the observed data. On this basis, we further evaluate the performance of the ZS14 scheme, and show that the accuracy of ZS14 scheme decreases as instability increases. As examples, Fig. 5b compared $V_{d,LES}$, of Exp (5, 9, 17) and Exp (24, 27, 33) with the ZS14 scheme result $V_{d,\tau_r}$ which is calculated using $\tau_r$. It shows that under weak wind conditions, $V_{d,\tau_r}$ predicts the deposition well under neutral conditions and underestimates the deposition under convective conditions, especially for particles that are not dominated by molecular diffusion and gravity, and the underestimation increases with the atmospheric instability. To predicts the deposition velocity more accurately for convective conditions, we need to account for the effect of shear-stress fluctuations, i.e., the instantaneous shear stress distribution. Thus, the dry deposition scheme can be improved as

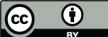

$$V_{d,\tau} = \int_0^\infty V_d(\tau) p(\tau) \, d\tau \tag{21}$$

with $p(\tau)$ is as given by Eqs. (18)-(20). As Fig. 5c shows, the improved scheme results $V_{d,\tau}$ and the simulation value $V_{d,LES}$ are shown a remarkable congruence.

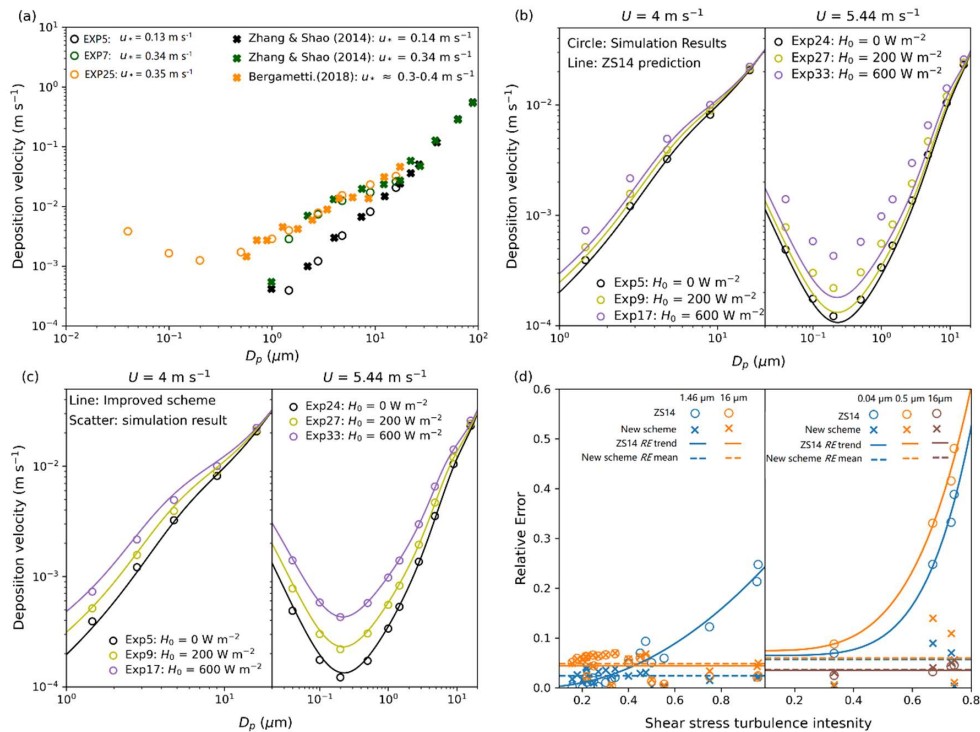

**Figure 5. (a)** Validation of the simulated deposition velocity from WRF-LES/D (circles) by comparing with the observation data (crosses). **(b)** the comparison of the predicted result by ZS14 scheme (lines) with the simulated value (circles) of Exp (5, 9, 17) (left) and Exp (24, 27, 33) (right). **(c)** the comparison of the predicted result by the improved scheme (lines) with the simulated value (circles) of Exp (5, 9, 17) (left) and Exp (24, 27, 33) (right). **(d)** Comparison of relative error as a function of shear stress turbulence intensity (TI-S), estimated by ZS14 scheme (circles) and the improved scheme (crosses) for Exp (1-20) (Left) and Exp (24, 27, 30, 33) (right).

To make the comparison more clear, the relative errors (*RE*) of the predicted deposition velocity by ZS14 scheme and improved scheme are compared with the WRF-LES/D simulation value and are calculated as below

$$RE = \left| \frac{V_{d,LES} - V_{d,\tau_r}\left(\text{or } V_{d,\tau}\right)}{V_{d,LES}} \right| \times 100\% \tag{22}$$

Analysis shows that the value of relative error, *RE*, depends on surface conditions, wind conditions, atmospheric stabilities, and particle sizes. It increases obviously with increased atmospheric instability under



weak wind conditions, while becomes less sensitive to stability when wind is strong. Through the analysis, we find that the *RE* of ZS14 scheme generally increases with the shear stress turbulence intensity, TI-S, and the value depends on particle size, as shown in Fig. 5d (left). Thus, we compared the *RE* of some different sized particles to investigate that the particle in which size range is strongly affected (Fig. A2). The result

shows that *RE* first increases and then decreases with increasing particle size, and the particles with size normally in the range of 0.01 to 10 are strongly affected by turbulent shear stress and $p(\tau)$ needs to be considered. After modification, the errors are limited to less or about 10%. For example, the relative error of Exp (17, i.e., $U = 4$ m s$^{-1}$ and $H_0 = 600$ W m$^{-2}$) for particles of 1.46 μm is reduced from ~ 25% to ~ 3%. The relative error of Exp (33, i.e., $U = 5.44$ m s$^{-1}$ and $H_0 = 600$ W m$^{-2}$) for particles of 0.5 μm is reduced from ~

50% to ~ 12%.

To further analyze if the *RE* of ZS14 in unstable conditions is dominated by kinetic instability or dynamic instability, the Richardson number is calculated. Analysis shows that TI-S is positively correlated to gradient Richardson number Ri, and *RE* of ZS14 is increasing with the magnitude of Richardson number Ri under convection predominant unstable condition associating weak winds and strong vertical motion (Fig. A3). The

relationship between Ri and TI-S needs further study. Consequently, the results illustrate that the modified scheme $V_{d,\tau}$ tends to be more accurate than the unmodified scheme $V_{d,\tau_r}$.

## 4. Conclusion

The present study was designed to determine the effect of ABL stability on dust particle deposition. For this purpose, the WRF-LES/D was used to model atmospheric boundary-layer turbulence under the presence of

atmospheric stability effects to recover statistics of shear stress variability. We then presented an improved dust-deposition scheme with the consideration of turbulent shear stress. While ABLS can broadly represent levels of atmospheric turbulence, its effect on dust deposition is wind speed dependent. Through a series of numerical experiments, we have shown the turbulent characteristics of dust deposition velocity caused by the turbulent wind flow and pointed out existing dust-deposition schemes have deficiencies in representing dust

deposition under convective conditions. The relative error *RE* increases as the ABL instability increases for low wind conditions, i.e., *RE* increases with shear stress turbulence intensity, especially for a certain size range of particles.

Since the dependency of dust deposition on micrometeorology is imbedded in the application of the surface shear stress, we believe that the dependency of dust deposition on ABL stability is ultimately attributed to

the statistical behavior of shear stress τ. Therefore, in this study, a model including the effects of surface shear fluctuations is proposed and validated by numerical experiments. Additionally, the fluctuations of surface shear caused by turbulence are available to estimate by a Weibull distribution function. The shape parameter decreases exponentially with the reciprocal of Monin-Obukhov length, and the scale parameter increases linearly with $u_{*r}^2 + 0.001 w_*^2$. After statistically revising the original scheme, an improved model is obtained.



Using the modified model, the deposition velocity tends towards numerical experimental results.

The project is the first comprehensive investigation of the turbulent characters of dust deposition and the findings will be of interest to improve the accuracy of dust-deposition predictions in regional or global scales. One source of weakness in this study is the variation of $\tau$ may change with surface roughness and needs further study. In spite of this limitation, the study adds to our understanding of the influence caused by ABLS

on particle deposition.

**Appendix**

Figure A1 shows the probability density distribution of surface shear stress for experiments (21-35); Figure A2 shows the changing of relative error with particle size; Figure A3 shows the variation of relative error (*RE*) of the ZS14 scheme (Eq. (10)) and improved scheme (Eq. (21)) with gradient Richardson number Ri.

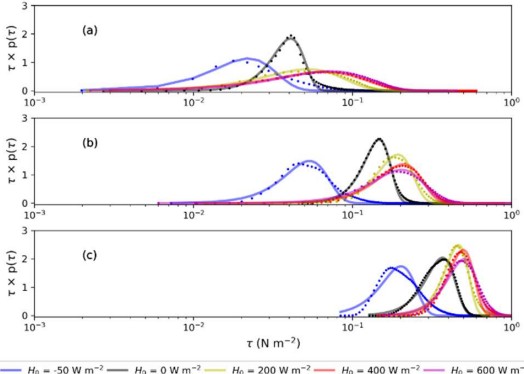


**Figure A1.** Probability distributions of simulated surface shear stress (dots) and the corresponding fitted Weibull density distribution (solid lines) with different surface heat flux for different wind conditions: **(a)** $U$ = 5.44 m s$^{-1}$, **(b)** $U$ = 10.87 m s$^{-1}$, **(c)** $U$ = 18.12 m s$^{-1}$.

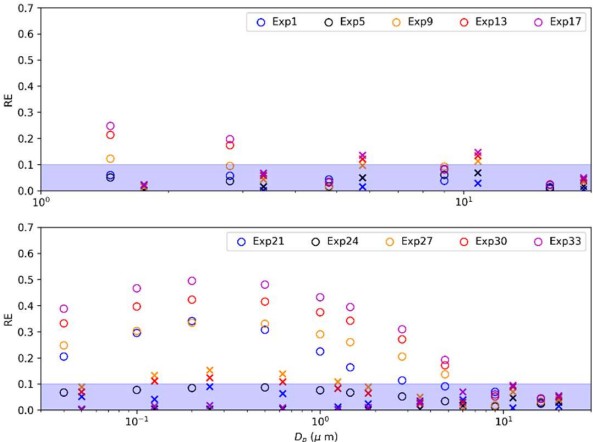





**Figure A2.** *RE* changes with particle size under weak wind conditions.

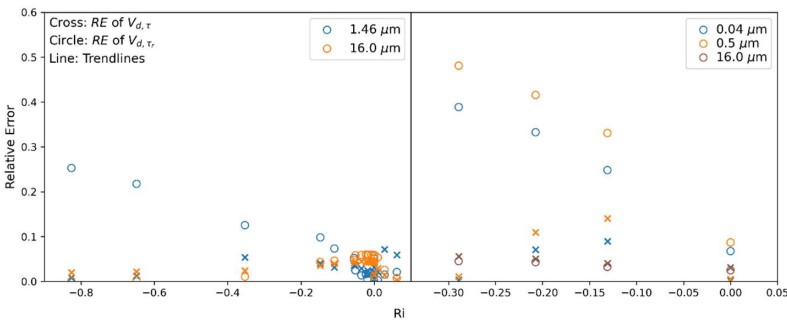

**Figure A3.** Comparison of relative error as a function of Ri, estimated by ZS14 scheme (circles) and the improved scheme (crosses) for Exp (1-20) (left) and Exp (24, 27, 30, 33) (right).

**Code and data availability**

The source code used in this study is the WRF-chem version 3.7 in the LES mode coupled with a new deposition scheme. WRF-LES model can be download at https://www2.mmm.ucar.edu/wrf/users/download/get_sources.html. The code of the coupled deposition scheme and data set obtained by the simulation are available online at https://github.com/YinXin2021/WRF-

LES-DustDepositionScheme.

**Author contributions**

XY, YPS and JZ were responsible for the formal analysis, Methodology. XY and CJ were responsible for the data curation, software, validation and visualization. YPS, JZ and NH were responsible for the supervision, project administration and funding acquisition. XY was responsible for investigation and Writing - original

draft preparation. XY, YPS and JZ were responsible for the Writing – review & editing.

**Competing interests**

The authors declare that they have no conflict of interest.

**Disclaimer**

Publisher's note: Copernicus Publications remains neutral with regard to jurisdictional claims in published

maps and institutional affiliations.



**Acknowledgments.**

We thank the Second Tibetan Plateau Scientific Expedition and Research Program (2019QZKK020611),the State Key Program of the National Natural Science Foundation of China (41931179), the China Scholarship Council (No. 201606180041) and the Fundamental Research Funds for the Central Universities (grant no. lzujbky-2020-cd06).

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
