# Peer review of "LES study on turbulent particle deposition and its dependence on atmospheric boundary-layer stability"

_Atmospheric Chemistry and Physics, 2021_

## Referee Comment (RC1)

This article addresses a well identified but poorly solved problem, namely the effect of atmospheric boundary layer stability (ABLS) on the dry deposition of particles. Indeed if it has been noted that the dry deposition velocities of particles are higher in unstable conditions, no convincing physical demonstration of the reasons for this increase in deposition velocities has been proposed and consequently no parameterization of this effect exists until now. This paper addresses this issue.

The idea behind this paper is based on the fact that much of the particle deposition is induced by turbulent diffusion which is related to turbulent shear stress and that turbulent shear stress (both its average value and its fluctuations) are related to the stability of the atmospheric boundary layer, with unstable conditions resulting in higher turbulent shear stress and, more importantly, larger fluctuations. Because the dry deposition velocity does not depend linearly on the shear stress and because existing models describe the deposition velocity only as a function of the average shear stress (or average friction velocity), the effect of fluctuations in the shear stress is not properly accounted for.

To address this question, the authors perform 35 runs of a sophisticated Large Eddy Simulation (LES) model corresponding to various stability, wind speed and roughness conditions. This approach is well adapted to the problem and the results are convincing and important.

The results clearly show that under unstable conditions, especially when the wind speed is low, the shear stress shows a strong variability around its mean value, whereas under neutral or stable conditions, with strong wind speed, this variability is strongly reduced. The authors then look at the consequences of these results on dry deposition velocities. The results are clear: in stable or neutral conditions, there is no impact of the stability on the calculated deposition velocities; these are identical whether or not the fluctuations in shear stress induced by the stability conditions are taken into account. On the other hand, for unstable conditions, the deposition velocities integrating the shear stress fluctuations are nearly 50% higher than those based on the average shear stress value are.

The demonstration is convincing and clearly points out that the turbulent fluctuations of the shear stress are responsible for the increase in deposition velocities under unstable conditions.

The authors will then develop a parameterization allowing to take into account this effect in the dry deposition modules. The first step is to define the probability distribution function of the shear stress. The authors demonstrate that this distribution can be correctly reproduced by Weibull distributions and determine a relation allowing the calculation of the parameters of these distributions as a function of, in particular, the Monin-Obukhov length and the average friction speed. Finally, they apply the improved dry deposition model and show that taking into account the effects of instability leads to a significant increase in the deposition velocity of particles between 0.1 and about 2 μm in size.

This paper is highly interesting and provides physical basis for the understanding of the effect of the ABLS on particle dry deposition. The paper is complete since it does not only explain the role played by ABLS but it proposes a parameterization that can be used into dry deposition modules to account for its effects. The paper is well written, concise, correctly illustrated with the correct references.

I think the impact of this article is greater than the title suggests. Indeed, the work developed concerns all types of particles and not only dusts. A title like *LES study on turbulent particles deposition and its dependence on atmospheric boundary-layer stability: application to dust* could better reflect the real content of the work. In any case, I highly recommend publishing this article in ACP with minor revisions.

Minor comments:

Line 24: Gregory (1945) is not in the reference list

Line 35: a review paper by Fowler et al. (Atmospheric Environment 43, 5193-5267, 2009) should be cited. I think that the authors could have benefit to have a look at paragraph 7.6.1.3 (understanding the effect of stability and leaf properties on deposition velocities, p5258).

Lines 39-40: the transition to dust is not well done, especially it is not clear why it is necessary to discuss about dust here. As mentioned before, and except if I miss something, the conclusions of the paper apply to any types of particles.

Line 81: TKE (Turbulent Kinetic Energy)

Line 88: $R_a$ is not defined

Line 112: change "grand" in "ground"

Line 146: How are the rebound and collection efficiency computed? At least add a reference

Line 226: 1-c and 1-f instead of 1a-c and 1d-f

Line 227: the choice of 1.46 µm for the particle size should be explained in few words

Line 242: Li et al., 2020 and not 2020a

Line 274: it should be useful for the reader to know how $1/L_o$ is computed in the simulation and also to give its value in table 2 (with only figure 4, it is not easy to connect $1/L_o$ to the other parameters)

Line 333: again I think that the conclusion is not specific to dust

Line 353: the authors mention that change in surface roughness may affect the variation of the shear stress. They have simulations for two different roughness lengths and have provided in Appendix a figure similar to figure 3 but for $z_o$=0.76 mm. Even, if the wind speed are not exactly the same, they could discuss a little more how change in roughness changes or not the pdf of shear stress.

Line 377: author contributions. Change YPS in YP to be consistent with the name of the authors used after the title.

Lines 419-420: delete this reference not cited in the text

Lines 435-436: delete this reference not cited in the text

Lines 443: change 2020a by 2020

Lines 444-445: this reference is in duplicate

---

## Author Comment (AC1)

To address this question, the authors perform 35 runs of a sophisticated Large Eddy Simulation (LES) model corresponding to various stability, wind speed and roughness conditions. This approach is well adapted to the problem and the results are convincing and important.

The results clearly show that under unstable conditions, especially when the wind speed is low, the shear stress shows a strong variability around its mean value, whereas under neutral or stable conditions, with strong wind speed, this variability is strongly reduced. The authors then look at the consequences of these results on dry deposition velocities. The results are clear: in stable or neutral conditions, there is no impact of the stability on the calculated deposition velocities; these are identical whether or not the fluctuations in shear stress induced by the stability conditions are taken into account. On the other hand, for unstable conditions, the deposition velocities integrating the shear stress fluctuations are nearly 50% higher than those based on the average shear stress value are.

The demonstration is convincing and clearly points out that the turbulent fluctuations of the shear stress are responsible for the increase in deposition velocities under unstable conditions.

The authors will then develop a parameterization allowing to take into account this effect in the dry deposition modules. The first step is to define the probability distribution function of the shear stress. The authors demonstrate that this distribution can be correctly reproduced by Weibull distributions and determine a relation allowing the calculation of the parameters of these distributions as a function of, in particular, the Monin-Obukhov length and the average friction speed. Finally, they apply the improved dry deposition model and show that taking into account the effects of instability leads to a significant increase in the deposition velocity of particles between 0.1 and about 2  $\mu$ m in size.

This paper is highly interesting and provides physical basis for the understanding of the effect of the ABLS on particle dry deposition. The paper is complete since it does not only explain the role played by ABLS but it proposes a parameterization that can be used into dry deposition modules to account for its effects. The paper is well written, concise, correctly illustrated with the correct references.

I think the impact of this article is greater than the title suggests. Indeed, the work developed concerns all types of particles and not only dusts. A title like LES study on turbulent particles deposition and its dependence on atmospheric boundary-layer stability: application to dust could better reflect the real content of the work. In any case, I highly recommend publishing this article in ACP with minor revisions.

Response: We are most grateful to Dr. Gilles Bergametti for his time and effort in reading the manuscript, as well as for his encouraging comments and insightful suggestions. These comments are very valuable for us to improve our paper and approach the truth. Dr. Gilles Bergametti pointed out the basic theory behind this study and the usefulness of this study in clarifying the dependence of turbulent particle deposition velocity on atmospheric boundary layer stability. We thank Dr. Gilles Bergametti for his comment that this work applies not only dusts but also other particles. Therefore, we will change the title to 'LES study on

turbulent particle deposition and its dependence on atmospheric boundary-layer stability' and modify the text accordingly.

**Minor comments:**

1) Line 24: Gregory (1945) is not in the reference list Response: *Thanks. We will correct it in the revision.*

2) Line 35: a review paper by Fowler et al. (Atmospheric Environment 43, 5193-5267, 2009) should be cited. I think that the authors could have benefit to have a look at paragraph 7.6.1.3 (understanding the effect of stability and leaf properties on deposition velocities, p5258).

Response: Thanks. Paragraph 7.6.1.3 in Fowler et al. (2009) noted the gaps in observations that can better control stability conditions and the lack of testable hypothesis explaining the link between dry deposition velocity and atmospheric stability. It helped us to better understand the influence of stability and vegetation properties on deposition velocities, and further research needs.

Line 35 will be revised to 'It has been observed that the dry deposition velocities under convective conditions are larger than those under neutral and stable conditions when the background wind speeds are similar, but there is no convincing physical scheme in models to account for the effects of the instability (Fowler et al., 2009).'

**3)** Lines 39-40: the transition to dust is not well done, especially it is not clear why it is necessary to discuss about dust here. As mentioned before, and except if I miss something, the conclusions of the paper apply to any types of particles.

Response: Sorry about this. We will replace the 'dust' using 'particle' in Lines 39-40, as well as in the conclusions part in the revision.

4) Line 81: TKE (Turbulent Kinetic Energy) Response: *Thanks. We will add the full name of TKE in the revision.*

5) Line 88:  $R_a$  is not defined Response: Sorry about this.  $R_a$  is the specific gas constant of air. We will define this in the revision.

6) Line 112: change "grand" in "ground" Response: *Thanks. We will correct this in the revision.*

7) Line 146: How are the rebound and collection efficiency computed? At least add a reference Response: *Thanks. We will give formulas and references for the rebound and collection efficiency in the revision.*

8) Line 226: 1-c and 1-f instead of 1a-c and 1d-f Response: *Thanks. We will correct this in the revision.*

9) Line 227: the choice of 1.46  $\mu$ m for the particle size should be explained in few words Response: WRF that we used calculates the deposition velocity by default for particles with diameters of 1.46  $\mu$ m, 2.8  $\mu$ m, 4.8  $\mu$ m, 9  $\mu$ m and 16  $\mu$ m. In the text, the particle size of 1.46  $\mu$ m is chosen as an example because the particle of this size is most sensitive to turbulent diffusion compared to the other four sizes.

10) Line 242: Li et al., 2020 and not 2020a Response: *Thanks. We will correct this in the revision.*

11) Line 274: it should be useful for the reader to know how  $1/L_o$  is computed in the simulation and also to give its value in table 2 (with only figure 4, it is not easy to connect  $1/L_o$  to the other parameters)

Response: Thanks for the suggestion.  $1/L_o$  is the reciprocal of the Monin-Obukhov length which is defined by  $L_o = \frac{-\overline{\theta}u_*^3}{kg\overline{w'\theta'_0}}$  with  $\overline{w'\theta'_0} = \frac{H_0}{\rho_a c_p}$ . Thus,  $1/L_o = -\frac{kg\overline{w'\theta'_0}}{\overline{\theta}u_*^3}$ . In the revision, we will give the computed formula of  $1/L_o$  in the text and its value in table 2.

12) Line 333: again I think that the conclusion is not specific to dust Response: *Thanks for the suggestion. We will correct it in the revision.*

13) Line 353: the authors mention that change in surface roughness may affect the variation of the shear stress. They have simulations for two different roughness lengths and have provided in Appendix a figure similar to figure 3 but for  $z_0=0.76$  mm. Even, if the wind speed are not exactly the same, they could discuss a little more how change in roughness changes or not the pdf of shear stress.

Response: The surface roughness length mainly reflects the fact that the surface topography changes the turbulence structure near the surface, which affects the mean wind profile. Since the simulations cannot delineate the surface topography in detail, we vary the roughness length to simulate different wind profile conditions. However, this does not fully reflect the effect of the change in surface topography on the turbulent structure and the particle deposition process under this effect. We thank the reviewer for this comment and we will examine this issue in detail in future work.

14) Line 377: author contributions. Change YPS in YP to be consistent with the name of the authors used after the title.

Response: Thanks. We will correct it in the revision.

15) Lines 419-420: delete this reference not cited in the text Response: *Thanks. We will delete this reference not cited in the text in the revision.*

16) Lines 435-436: delete this reference not cited in the text Response: *Thanks. We will correct this in the revision.*

17) Lines 443: change 2020a by 2020 Response: *Thanks. We will correct this in the revision.*

18) Lines 444-445: this reference is in duplicate Response: *Thanks. We will delete this in the revision.*

---

## Author Comment (AC2)

**Response to anonymous referee #2's interactive comment on the manuscript "LES study on turbulent dust deposition and its dependence on atmospheric boundary-layer stability"**

**General comments:** The authors use large-eddy simulation (LES) to investigate the influence of turbulent shear stress / momentum flux on dust deposition. Using a shear-stress weighted average of dry deposition velocity, they derive a modified version of a dust deposition scheme and obtain improved results compared to the LES dust deposition.
The subject of investigation is important and tests of the impact of the improved parameterization on the spatial distribution of dust deposition in a regional or global model are desirable in a future study. The manuscript is well structured. I therefore recommend publication of the manuscript after consideration of the following comments, which are overall minor.

Response: We are most grateful to Referee #2 for the time and effort he/she put into reading the manuscript, and for his/herhelpful comments and constructive suggestions. We fully agree with the Referee #2's suggestion about further investigation and tests of the improved parameterization of dust deposition in a regional or global model. There are several suggestions we will consider and modify the text accordingly.

**Minor comments:**
1) How is dust deposition (velocity and fluxes) calculated in the LES? Is the deposition scheme from Zhang and Shao (2014) used here as well? This does not become clear in the text.
Response: *Yes, the deposition scheme from Zhang and Shao (2014) is used for each grid of the lowest layer in the LES to calculate the deposition flux on the ground. In the revision, we will add a line of explanation to make it clear.*

2) Apart from the two different roughness lengths, the LES simulation design (domain configuration, simulation setup, cases) seem to be exactly as in Klose and Shao (2013), as are components of the analysis of the shear stress distribution. It should be made clear in the text that parts of the study design follow Klose and Shao (2013).
Response: *Sorry about this. We will try to clarify that parts of this study design follow Klose and Shao (2013).*

3) Data used in the paper is made available online, which is great. Ideally, I think a format which is independent of the programming language used would be preferable. Currently npy is used, which requires python. This is only a recommendation.
Response: *Thanks for the suggestion. We will try to convert the data from npy format to a csv format that is independent of the programming language.*

4) Line 9-10 While there are studies on the effects of atmospheric boundary layer stability (ABLS) on dust emission, I do not agree that they are as clearly documented as the sentence suggests. Stability is not typically considered in dust emission schemes. I propose to revise the sentence.
Response: *Indeed, this sentence is not accurately expressed. We will revise the sentence to 'While the effects of ABLS on particle emission have received more attention, those on particle deposition have rarely been explored' in the revision.*

5) Line 26 When stating that several dust deposition schemes have been proposed, I recommend listing more than two.
Response: *Thanks for the suggestion. We will give more dust deposition schemes in the revision.*

6) Line 43-44 Please add reference.
Response: *Thanks. We will add the reference in the revision.*

7) Line 50-53 This is (almost entirely) a direct citation from Klose and Shao (2013) and should be indicated as such.
Response: *Thanks. We will indicate this in the revision.*

8) Line 56-57 Sentence (current dust-deposition schemes only consider the mean wind) needs reference.
Response: *Thanks. We will correct this in the revision.*

9) Line 60 to accurately model
Response: *Thanks. We will correct it in the revision.*

10) Line 77 What do you mean with "reasonably well-established"?
Response: *Indeed, this is a bit sloppy. We will try to be more precise in the revision.*

11) Line 82 "nonlinear backscatter and anisotropic" – please check grammar
Response: *Thanks. We will check and correct it in the revision.*

12) Line 92-95 I presume the description of tau_ij is inherent to WRF, in which case a reference should be added.
Response: *Thanks. We will list the reference in the revision.*

13) Line 99 divided by
Response: *Thanks. We will correct it in the revision.*

14) Line 107 where K_m is eddy viscosity and phi_m is the MOST stability function
Response: *Thanks. We will correct it in the revision.*

15) Line 112 with "on grand" do you mean grid-resolved or grid-scale?
Response: *This comment is similar to Referee 1's comment. The 'grand' will be changed to 'ground' in the revision.*

16) Line 114/115 as the change of dust concentration close to the surface
Response: *Thanks. We will correct it in the revision.*

17) Line 117 The combination of the two references given for dust emission is a little confusing, as the Shao (2004) paper deals with a dust emission scheme (without consideration of turbulence effects) and the paper from Klose and Shao (2013) deals with turbulent dust emission, but is no dust emission scheme (the corresponding references would be Klose and Shao (2012) and Klose et al. (2014)). Please clarify what the intention is here and update the references accordingly.
Response: *Indeed, this is a bit confusing. The purpose of the combination is to show the dust emission schemes have been studied with and without considering turbulence effects. We will try to be more precise and change Klose and Shao (2013) to Klose and Shao (2012) and Klose et al. (2014) accordingly.*

18) Line 118 settling instead of settlement
Response: *Thanks. We will correct it.*

19) Line 134 r_g should be defined at its first occurrence directly after Equation 10.
Response: *Thanks. We will correct it in the revision.*

20) Line 137 Please indicate the particle-size regime for which the Stokes assumption of a linear dependence of drag coefficient on particle Reynolds number, which is used here, is appropriate.
Response: Thanks. *According to* Malcolm and Raupach (1991), *the Stokes regime is restricted to $D_p < 20$ μm for quartz spheres falling freely in the air. We will give the corresponding particle-size regime in the revision.*

21) Line 141 assumption that dust concentration is zero
Response: *Thanks. We will correct it in the revision.*

22) Line 156 with beta … being the ratio…
Response: *Thanks. We will correct it in the revision.*

23) Line 158 for particles with diameter
Response: *Thanks. We will correct it in the revision.*

24) Line 160 gravitational settling
Response: *Thanks. We will correct it in the revision.*

25) Line 189 remove "below"
Response: *Thanks. We will correct it in the revision.*

26) Line 207 Do you mean "decreases with increasing wind speed"?
Response: *Sorry about this. Indeed, this line is a bit confusing. As Referee #2 pointed out, we had hoped to convey that 'decreases with increasing wind speed'. We will correct it in the revision.*

27) Line 211 ABLs, buoyancy
Response: *Thanks. We will correct it in the revision.*

28) Line 226 fluctuating behavior
Response: *Thanks. We will correct it in the revision.*

29) Line 287 performance
Response: *Thanks. We will correct it in the revision.*

30) Line 291-292 Check grammar
Response: *Thanks for Referee #2's kind reminder. We will change Line 291-292 to 'On this basis, by further evaluating the performance of the scheme of ZS14, we found that the accuracy of the ZS14 scheme decreases with increasing instability. As examples, we compared the deposition velocities $V_{d,LES}$ from Exp (5,9,17) and Exp (24, 27,33) with the deposition velocities $V_{d,\tau_r}$ calculated by the ZS14 scheme using each corresponding $\tau_r$.' in the revision.*

31) Line 295 To predict
Response: *Thanks. We will correct it in the revision.*

32) Fig. 5 Check and use consistent labels (e.g. scatter/line versus circle/line)
Response: *Thanks. We will check and correct the inconsistent labels.*

33) Line 316 while it becomes
Response: *Thanks. We will correct it.*

34) Line 326 Please state how you calculated the Richardson number.
Response: *The Richardson number is derived from the transformed form of the formula $R_i = \frac{g}{\theta}\frac{\partial \overline{\theta}}{\partial z}\left(\frac{\partial V}{\partial z}\right)^{-2}$. According to Li et al. (2014), the formula can be rewritten as $R_i = -\frac{kz\varphi_h}{z_i\varphi_m^2}\left(\frac{w_*}{u_*}\right)^3$. As $w_* = \left(\frac{g}{\theta}\overline{w'\theta'}_0 z_i\right)^{1/3}$, $R_i$ can be estimated by using $R_i = -\frac{g}{\theta}kz\frac{\varphi_h}{\varphi_m^2}\frac{\overline{w'\theta'}_0}{u_*^3}$. In the equation, z used is the center height of the lowest layer in this study, $\varphi_h$ and $\varphi_m$ are the Monin-Obukhov similarity functions*

*for heat and momentum, respectively, which can be calculated by referring to Shao (2009), and* $\overline{w'\theta'}_0 = \dfrac{H_0}{\rho_a c_p}$. *We will give the calculated formula in the revision.*

35) Line 328 associated with
Response: *Thanks. We will correct it in the revision.*

36) Line 338 In principle the deficiencies have only been shown for one dust-deposition scheme, even though from a conceptual point of view, this means that it applies for other schemes as well. I suggest to rephrase the sentence slightly to account for this nuance.
Response: *Thanks for the suggestion and we agree with Referee #2. We will modify this sentence to 'Through a series of numerical experiments, we have shown the turbulent characteristics of dust particle deposition velocity caused by the turbulent wind flow and pointed out the scheme of ZS14 has deficiencies in representing particle deposition under convective conditions.'*

37) Line 342 embedded
Response: *Thanks. We will correct it in the revision.*

38) Line 346 can be approximated with a Weibull distribution
Response: *Thanks. We will correct it in the revision.*

39) Line 351 on regional or global scales
Response: *Thanks. We will correct it in the revision.*

40) Line 352 the variation of tau may be changed (or affected) by surface roughness
Response: *Thanks. We will correct it in the revision.*

**References**

*Klose, M. and Shao, Y.: Stochastic parameterization of dust emission and application to convective atmospheric conditions, Atmos. Chem. Phys., 12(16), 7309–7320, doi:10.5194/acp-12-7309-2012, 2012.*

*Klose, M. and Shao, Y.: Large-eddy simulation of turbulent dust emission, Aeolian Res., 8, 49–58, doi:10.1016/j.aeolia.2012.10.010, 2013.*

*Klose, M., Shao, Y., Li, X., Zhang, H., Ishizuka, M., Mikami, M. and Leys, J. F.: Further development of a parameterization for convective turbulent dust emission and evaluation based on field observations, J. Geophys. Res., 119(17), 10441–10457, doi:10.1002/2014JD021688, 2014.*

*Li, X. L., Klose, M., Shao, Y. and Zhang, H. S.: Convective turbulent dust emission (CTDE) observed over Horqin Sandy Land area and validation of a CTDE scheme, J. Geophys. Res. Atmos., 119, 9980–9992, doi:10.1002/2014JD021572.Received, 2014.*

*Malcolm, L. P. and Raupach, M. R.: Measurements in an air settling tube of the terminal velocity distribution of soil material, J. Geophys. Res., 96, 15,275-15,286, 1991.*

*Shao, Y.: Physics and Modelling of Wind Erosion., Springer Verlag, 2008.*

*Zhang, J. and Shao, Y.: A new parameterization of particle dry deposition over rough surfaces, Atmos. Chem. Phys., 14(22), 12429–12440, doi:10.5194/acp-14-12429-2014, 2014.*

---

## Author Response (AR1)

To address this question, the authors perform 35 runs of a sophisticated Large Eddy Simulation (LES) model corresponding to various stability, wind speed and roughness conditions. This approach is well adapted to the problem and the results are convincing and important.

The results clearly show that under unstable conditions, especially when the wind speed is low, the shear stress shows a strong variability around its mean value, whereas under neutral or stable conditions, with strong wind speed, this variability is strongly reduced. The authors then look at the consequences of these results on dry deposition velocities. The results are clear: in stable or neutral conditions, there is no impact of the stability on the calculated deposition velocities; these are identical whether or not the fluctuations in shear stress induced by the stability conditions are taken into account. On the other hand, for unstable conditions, the deposition velocities integrating the shear stress fluctuations are nearly 50% higher than those based on the average shear stress value are.

The demonstration is convincing and clearly points out that the turbulent fluctuations of the shear stress are responsible for the increase in deposition velocities under unstable conditions.

The authors will then develop a parameterization allowing to take into account this effect in the dry deposition modules. The first step is to define the probability distribution function of the shear stress. The authors demonstrate that this distribution can be correctly reproduced by Weibull distributions and determine a relation allowing the calculation of the parameters of these distributions as a function of, in particular, the Monin-Obukhov length and the average friction speed. Finally, they apply the improved dry deposition model and show that taking into account the effects of instability leads to a significant increase in the deposition velocity of particles between 0.1 and about 2  $\mu$ m in size.

This paper is highly interesting and provides physical basis for the understanding of the effect of the ABLS on particle dry deposition. The paper is complete since it does not only explain the role played by ABLS but it proposes a parameterization that can be used into dry deposition modules to account for its effects. The paper is well written, concise, correctly illustrated with the correct references.

I think the impact of this article is greater than the title suggests. Indeed, the work developed concerns all types of particles and not only dusts. A title like LES study on turbulent particles deposition and its dependence on atmospheric boundary-layer stability: application to dust could better reflect the real content of the work. In any case, I highly recommend publishing this article in ACP with minor revisions.

Response: We are most grateful to Dr. Gilles Bergametti for his time and effort in reading the manuscript, as well as for his encouraging comments and insightful suggestions. These comments are very valuable for us to improve our paper and approach the truth. Dr. Gilles Bergametti pointed out the basic theory behind this study and the usefulness of this study in clarifying the dependence of turbulent particle deposition velocity on the atmospheric boundary layer stability. We thank Dr. Gilles Bergametti for pointing out that this work applies not only to dust but also to other particles.

Therefore, we changed the title to "LES study on turbulent particle deposition and its dependence on atmospheric boundary-layer stability" and modify the text accordingly.

**Minor comments:**

1) Line 24: Gregory (1945) is not in the reference list

Response: Thanks. we corrected this same as below in the revision. Gregory, P. H.: The dispersion of airborne spores, Trans. British Mycological Soc., 28(1-2), 26–72, doi:10.1016/s0007-1536(45)80041-4, 1945.

2) Line 35: a review paper by Fowler et al. (Atmospheric Environment 43, 5193-5267, 2009) should be cited. I think that the authors could have benefit to have a look at paragraph 7.6.1.3 (understanding the effect of stability and leaf properties on deposition velocities, p5258).

Response: Thanks. In the revision, the review paper by Fowler et al. (2009) was cited. In addition, paragraph 7.6.1.3 in Fowler et al. (2009) helps us to better understand the influence of stability and leaf properties on deposition velocities. This paragraph points out that there is no testable hypothesis in current models explaining the link between increasing deposition velocity and atmospheric stability. Furthermore, it tells us the leaf properties may affect deposition as the morphology and distribution density of epicuticular waxes significantly affect their hydrophobicity and anti-adhesion properties and potentially the adhesion of aerosol particles following impaction and interception. It helped us to better understand the influence of stability and vegetation properties on deposition velocities, and further research needs.

3) Lines 39-40: the transition to dust is not well done, especially it is not clear why it is necessary to discuss about dust here. As mentioned before, and except if I miss something, the conclusions of the paper apply to any types of particles.

Response: Thanks. We agree with your point. This sentence was corrected as below in the revision. "Some aeolian processes, e.g., turbulent particle emission (Klose and Shao, 2012, 2013) and intermittent saltation (Liu et al., 2018; Li et al., 2020; Rana et al., 2020), have been under development. To the best of our knowledge, although turbulent particle deposition is now perceived to be important, a scheme is yet to be constructed for its quantitative estimate."

4) Line 81: TKE (Turbulent Kinetic Energy)

Response: Thanks. The full name of TKE was added in the revision.

5) Line 88:  $R_a$  is not defined Response: *Thanks*.  $R_a$  is the specific gas constant of air. We corrected this in the revision.

6) Line 112: change "grand" in "ground" Response: *Thanks. We changed the word "grand" using "ground" in the revision.*

7) Line 146: How are the rebound and collection efficiency computed? At least add a reference Response: *Thanks. We added the computing formulas for rebound and collection efficiency in the revision as below.*

" $E_B$ ,  $E_{im}$ ,  $E_{in}$ , and R are expressed as

$$E_B = C_B S_c^{-2/3} \operatorname{Re}^{n_B - 1}$$
(14)

$$E_{im} = \left(\frac{S_t}{0.6 + S_t}\right)^2 \tag{15}$$

$$E_{in} = u_* 10^{-S_i} \frac{2D_p}{d_c}$$
(16)

$$R = \exp(-\sqrt{S_t}) \tag{17}$$

where Re is the roughness element Reynolds number,  $C_B$  and  $n_B$  are parameters depending on Re, and  $d_c$  is the diameter of the roughness element, and  $S_t$  is the Stokes number."

8) Line 226: 1-c and 1-f instead of 1a-c and 1d-f

Response: Thanks. We corrected this in the revision.

9) Line 227: the choice of 1.46 µm for the particle size should be explained in few words

Response: Thanks. Exp (1-20) used the default particle sizes (1.46, 2.8, 4.8, 9 and 16  $\mu$ m) of WRF-LES/D. In the text, the particle size of 1.46  $\mu$ m is chosen as an example because this size is the most sensitive to turbulent diffusion compared to the other four sizes. We explained it in the revised manuscript as follow:

"This size is chosen because it is the most sensitive to turbulent diffusion compared to the other four sizes (2.8, 4.8, 9, 16  $\mu$ m) used in Exp (1-20)."

10) Line 242: Li et al., 2020 and not 2020a

Response: Thanks. We corrected this in the revision.

11) Line 274: it should be useful for the reader to know how  $1/L_o$  is computed in the simulation and also to give its value in table 2 (with only figure 4, it is not easy to connect  $1/L_o$  to the other parameters)

Response: Thanks for the suggestion.  $1/L_o$  is the reciprocal of the Monin-Obukhov length which is defined by  $L_o = \frac{-\overline{\Theta}u_*^3}{k\overline{gw'\Theta'_0}}$  with  $\overline{w'\Theta'_0} = \frac{H_0}{\rho_a c_p}$ . Thus,  $1/L_o = -\frac{k\overline{gw'\Theta'_0}}{\overline{\Theta}u_*^3}$ . In the revision, we gave the computing formula of  $L_o$  same as below in the text and the value of  $1/L_o$  in table 2.

" $|1/L_0|$  is the absolute value of the reciprocal of the Obukhov length  $L_0$  that can be calculated using

$$L_o = \frac{-\overline{\partial}u_*^3}{kg\frac{H_0}{\rho_a c_p}}$$
(23)"

12) Line 333: again I think that the conclusion is not specific to dust

Response: Thanks. We corrected the sentence the same as below in the revision. "The present study was designed to determine the effect of ABL stability on particle deposition."

13) Line 353: the authors mention that change in surface roughness may affect the variation of the shear stress. They have simulations for two different roughness lengths and have provided in Appendix a figure similar to figure 3 but for  $z_0=0.76$  mm. Even, if the wind speed are not exactly the same, they could discuss a little more how change in roughness changes or not the pdf of shear stress.

Response: Thanks. The surface roughness length mainly reflects the fact that the surface topography changes the turbulence structure near the surface, which affects the mean wind profile. Since the

simulations cannot delineate the surface topography in detail, we vary the roughness length to simulate different wind profile conditions. However, this does not fully reflect the effect of the change in surface topography on the turbulence structure and the particle deposition process under this effect. We thank the reviewer for this comment and we will examine this issue in detail in future work.

14) Line 377: author contributions. Change YPS in YP to be consistent with the name of the authors used after the title.

Response: Thanks. It was corrected in the revision.

15) Lines 419-420: delete this reference not cited in the text

Response: Thanks. It was deleted in the revision.

16) Lines 435-436: delete this reference not cited in the text

Response: Thanks. It was deleted in the revision.

17) Lines 443: change 2020a by 2020

Response: Thanks. It was corrected in the revision.

18) Lines 444-445: this reference is in duplicate

Response: Thanks. It was corrected in the revision.

**Referee #2**

Response to anonymous referee #2's interactive comment on the manuscript "LES study on turbulent dust deposition and its dependence on atmospheric boundary-layer stability"

**General comments:** The authors use large-eddy simulation (LES) to investigate the influence of turbulent shear stress / momentum flux on dust deposition. Using a shear-stress weighted average of dry deposition velocity, they derive a modified version of a dust deposition scheme and obtain improved results compared to the LES dust deposition.

The subject of investigation is important and tests of the impact of the improved parameterization on the spatial distribution of dust deposition in a regional or global model are desirable in a future study. The manuscript is well structured. I therefore recommend publication of the manuscript after consideration of the following comments, which are overall minor.

Response: We are most grateful to Referee #2 for the time and effort he/she put into reading the manuscript, and for his/her helpful comments and constructive suggestions. We fully agree with Referee #2's suggestion about further investigation and tests of the improved parameterization of dust deposition in a regional or global model. There are several suggestions we have considered and modified the text accordingly.

**Minor comments:**

1) How is dust deposition (velocity and fluxes) calculated in the LES? Is the deposition scheme from Zhang and Shao (2014) used here as well? This does not become clear in the text.

Response: Thanks. In the LES, the deposition scheme from Zhang and Shao (2014) is used to calculate the deposition flux to the ground. In the last part of section 2.1 of the revised version, we gave the following description to make it clear:

"The surface heat flux, denoted  $H_0$ , is specified. The dry deposition flux to the ground for each grid, denoted  $F_d$ , is obtained by multiplying the deposition velocity  $V_d$  and particle concentration c in the lowest layer, and  $V_d$  is estimated using the ZS14 deposition scheme."

2) Apart from the two different roughness lengths, the LES simulation design (domain configuration, simulation setup, cases) seem to be exactly as in Klose and Shao (2013), as are components of the analysis of the shear stress distribution. It should be made clear in the text that parts of the study design follow Klose and Shao (2013).

Response: Sorry about this. In the revision, we added one sentence in the last paragraph of the introduction part to clarify that parts of this study design follow Klose and Shao (2013), as shown below:

"A large-eddy simulation (LES) model is used here to simulate turbulence and particle deposition under various ABLS conditions, and parts of the study design follow Klose and Shao (2013)."

3) Data used in the paper is made available online, which is great. Ideally, I think a format which is independent of the programming language used would be preferable. Currently npy is used, which requires python. This is only a recommendation.

Response: Thanks for the suggestion. We converted the data from npy format to a csv format that is independent of the programming language.

4) Line 9-10 While there are studies on the effects of atmospheric boundary layer stability (ABLS) on dust emission, I do not agree that they are as clearly documented as the sentence suggests. Stability is not typically considered in dust emission schemes. I propose to revise the sentence.

Response: Thanks. Indeed, this sentence was not accurately expressed. We revised the sentence to 'While the effects of ABLS on particle emission have attracted much attention and been investigated in several studies, those on particle deposition are so far less-well studied.' in the revision.

5) Line 26 When stating that several dust deposition schemes have been proposed, I recommend listing more than two.

Response: Thanks for the suggestion. In the revised manuscript, we listed four particle-deposition schemes, as shown below:

"Several particle-deposition schemes have been proposed (Slinn, 1982; Walcek et al., 1986; Zhang and Shao, 2014; Zhang et al., 2001)"

6) Line 43-44 Please add reference.

Response: Thanks. The reference was added in the revised version, same as below. "The turbulent wind flow in the particle-deposition schemes is reflected in the turbulent shear stress (or vertical momentum flux) (Fowler et al., 2009; Zhang and Shao, 2014)."

7) Line 50-53 This is (almost entirely) a direct citation from Klose and Shao (2013) and should be indicated as such.

Response: Thanks. This was corrected the same as below in the revision. Klose and Shao (2013) pointed out that:

In a convective atmospheric boundary layer, large eddies have coherent structures of dimensions comparable to boundary-layer depth. These eddies are efficient entities in generating localized momentum fluxes to the surface. Although the eddies only occupy fractions of time and space, the momentum fluxes to these fractions can be many times the average. (p. 49)

8) Line 56-57 Sentence (current dust-deposition schemes only consider the mean wind) needs reference.

Response: Thanks. It was corrected in the revision, same as below. "The current particle-deposition schemes only consider the mean behavior of wind (e.g., Slinn, 1982; Zhang and Shao, 2014; Zhang et al., 2001)"

9) Line 60 to accurately model

Response: Thanks. It was corrected in the revision:

10) Line 77 What do you mean with "reasonably well-established"?

Response: Thanks. Indeed, this is a bit sloppy. We removed "reasonably" in the revision, and the sentence becomes:

"As demonstrated in the earlier studies, WRF-LES/D is a well-established system for applications to simulating turbulence, turbulent particle emission and transport for various ABLS conditions."

11) Line 82 "nonlinear backscatter and anisotropic" - please check grammar

Response: Thanks. We changed the sentence the same as below in the revision. "The k-l subgrid closure (Deardorff, 1980) together with the TKE (Turbulent Kinetic Energy) equation (Skamarock et al., 2008) is used here."

12) Line 92-95 I presume the description of tau\_ij is inherent to WRF, in which case a reference should be added.

Response: Thanks. It was corrected the same as below in the revision.

" $\tau_{ij}$  is the subgrid stress tensor modeled using an eddy viscosity approach where the eddy viscosity is represented as the product of a length scale and a velocity scale characterizing the subgrid-scale (SGS) turbulent eddies (Dupont et al., 2013), with the velocity scale being derived from the SGS TKE and the length scale from the grid spacing (Skamarock et al., 2008)."

13) Line 99 divided by

Response: Thanks. It was corrected in the revision.

14) Line 107 where K\_m is eddy viscosity and phi\_m is the MOST stability function Response: Thanks. We corrected it in the revision according to Referee's comment, same as below. "where  $K_m$  is the eddy viscosity and  $\varphi_m$  is the MOST stability function"

15) Line 112 with "on grand" do you mean grid-resolved or grid-scale?

Response: Thanks. This comment is similar to Referee 1's comment. The word was misspelled and should have been "ground" instead of "great". In the revised version, the sentence is rewritten as "Furthermore, the surface heat flux, denoted H0, is specified. The dry deposition flux to the ground for each grid, denoted as Fd, is obtained by multiplying the deposition velocity Vd and particle concentration c in the lowest layer, and Vd is estimated using the ZS14 deposition scheme."

16) Line 114/115 as the change of dust concentration close to the surface

Response: Thanks. It was corrected in the revision, same as below. "The particle deposition on the surface is more complicated than momentum flux as the change of particle concentration close to the surface is unclear."

17) Line 117 The combination of the two references given for dust emission is a little confusing, as the Shao (2004) paper deals with a dust emission scheme (without consideration of turbulence effects) and the paper from Klose and Shao (2013) deals with turbulent dust emission, but is no dust emission scheme (the corresponding references would be Klose and Shao (2012) and Klose et al. (2014)). Please clarify what the intention is here and update the references accordingly.

Response: Thanks. Indeed, this is a bit confusing. The purpose of the combination is to show the dust emission schemes have been studied with and without considering turbulence effects. In the revision, we revised the sentence the same as below.

"The problem of particle emission has been dealt with elsewhere (e.g., Shao (2004) focuses on dust emission without turbulence effects; Klose et al., (2014) and Klose and Shao (2012) emphasize the turbulent particle emission) and is not considered here."

18) Line 118 settling instead of settlement

Response: Thanks. It was corrected in the revision.

19) Line 134 r g should be defined at its first occurrence directly after Equation 10.

Response: Thanks. We corrected it by defining  $r_g$  directly after the equation in the revision, as shown below.

$${}^{\prime\prime}V_d(z) = \left(r_g + \frac{r_s - r_g}{\exp(r_a / r_g)}\right)^{-1}$$
(10)

with  $r_g$  being the gravitational resistance,  $r_s$  being the collection resistance, and  $r_a$  being the aerodynamic resistance for the inertial layer.

The gravitational resistance  $r_g$  is defined as the reciprocal of the gravitational settling velocity  $w_t$  and depends mainly on particle size and density. A free-falling particle is subject to gravitational and aerodynamic drag forces. When these forces are in equilibrium, the gravitational settling velocity of the particle smaller than 20  $\mu$ m can be reasonably accurately calculated according to the Stokes formula (Malcolm and Raupach, 1991; Seinfeld and Pandis, 2006).

$$w_{t} = \frac{C_{u}\rho_{p}D_{p}^{2}g}{18\mu_{a}} = r_{g}^{-1}$$
(11)

where  $D_p$  is the particle diameter,  $\rho_p$  is the particle density,  $\mu_a$  is the air dynamic viscosity,  $C_u$  is the Cunningham correction factor that accounts for the slipping effect affecting the fine particles. Using the MOST, the aerodynamic resistance is calculated as

$$r_{a} = \frac{S_{cT}}{ku_{*}} \left[ \ln \left( \frac{z - z_{d}}{h - z_{d}} \right) - \psi_{m} \right]$$
(12)

where  $z_d$  is the displacement height, h is the height of roughness element  $\psi_m$  is the integral of stability function in the inertial layer,  $S_{cT} = K_m/K_p$  (Csanady, 1963), and  $\kappa$  is the von Karman constant."

20) Line 137 Please indicate the particle-size regime for which the Stokes assumption of a linear dependence of drag coefficient on particle Reynolds number, which is used here, is appropriate.

Response: Thanks. With reference to Malcolm and Raupach (1991) and Seinfeld and Pandis (2006), this Stokes assumption is accurate for a particle with size  $Dp

Response: Thanks. In the revision, the text in the plots was corrected to be consistent with the labels, as shown below.

*Figure 5. (a)* Validation of the simulated deposition velocity from WRF-LES/D (circles) by comparing with the observation data (crosses). (b) the comparison of the predicted result by ZS14 scheme (lines) with the simulated value (circles) of Exp (5, 9, 17) (left) and Exp (24, 27, 33) (right). (c) the comparison of the predicted result by the improved scheme (lines) with the simulated value (circles) of Exp (5, 9, 17) (left) and Exp (24, 27, 33) (right). (d) Comparison of relative error as a function of shear stress turbulence intensity (TI-S), estimated by ZS14 scheme (circles) and the improved scheme (crosses) for Exp (1-20) (Left) and Exp (24, 27, 30, 33) (right).

**33) Line 316 while it becomes**

**Response: Thanks. it was corrected in the revision.**

34) Line 326 Please state how you calculated the Richardson number.

Response: Thanks. The Richardson number is derived from the formula  $R_i = \frac{g}{\overline{\theta}} \frac{\partial \overline{\theta}}{\partial z} \left(\frac{\partial V}{\partial z}\right)^{-2}$ , where  $\partial \overline{\theta} = -\frac{\overline{\theta}' w'}{\partial z} = -\frac{\partial V}{\partial z} \left(\frac{\partial V}{\partial z}\right)^{-2}$ , where  $\partial \overline{\theta} = -\frac{\overline{\theta}' w'}{\partial z} = -\frac{\partial V}{\partial z} \left(\frac{\partial V}{\partial z}\right)^{-2}$ .

 $\frac{\partial \overline{\theta}}{\partial z} = -\frac{\overline{\theta'w'}}{ku_*z/\phi_h} and \frac{\partial V}{\partial z} = \frac{u_*}{kz/\phi_m}, \ \varphi_h \ and \ \varphi_m \ are \ the \ Monin-Obukhov \ similarity \ functions \ for \ heat \ and \ momentum, \ respectively, \ which \ can \ be \ calculated \ with \ reference \ to \ Shao \ (2008). \ Thus, \ the \ and \$

Richardson number can be estimated by using  $R_i = -\frac{g}{\theta}kz\frac{\varphi_h}{\varphi_m^2}\frac{1}{u_*^3}\frac{H_0}{\rho_a c_p}$ , where z is the center height of the lowest layer in this study. The formula of Ri was given in the appendix in the revision.

35) Line 328 associated with

Response: Thanks. It was corrected in the revision, same as below. "Under unstable conditions associated with strong vertical motion and weak winds, RE of ZS14 increases with the increasing magnitude of Richardson number Ri"

36) Line 338 In principle the deficiencies have only been shown for one dust-deposition scheme, even though from a conceptual point of view, this means that it applies for other schemes as well. I suggest to rephrase the sentence slightly to account for this nuance.

Response: Thanks for the suggestion and we agree with Referee #2. In the revision, we modified this sentence to

'Through a series of numerical experiments, we have shown the turbulent characteristics of particle deposition velocity caused by the turbulent wind flow and pointed out the shortcomings of the ZS14 scheme in representing particle deposition under convective conditions.'

37) Line 342 embedded

Response: Thanks. It was corrected in the revision.

38) Line 346 can be approximated with a Weibull distribution

Response: Thanks. It was corrected in the revision.

39) Line 351 on regional or global scales

Response: Thanks. It was corrected in the revision.

40) Line 352 the variation of tau may be changed (or affected) by surface roughness

Response: Thanks. It was corrected in the revision.

**References**

Csanady, G. T.: Turbulent Diffusion of Heavy Particles in the Atmosphere, J. Atmos. Sci., 20, 201–208, doi:10.1175/1520-0469(1964)021<0322:codohp>2.0.co;2, 1963.

Fowler, D., Pilegaard, K., Sutton, M. A., Ambus, P., Raivonen, M., Duyzer, J., Simpson, D., Fagerli, H., Fuzzi, S., Schjoerring, J. K., Granier, C., Neftel, A., Isaksen, I. S. A., Laj, P., Maione, M., Monks, P. S., Burkhardt, J., Daemmgen, U., Neirynck, J., Personne, E., Wichink-Kruit, R., Butterbach-Bahl, K., Flechard, C., Tuovinen, J. P., Coyle, M., Gerosa, G., Loubet, B., Altimir, N., Gruenhage, L., Ammann, C., Cieslik, S., Paoletti, E., Mikkelsen, T. N., Ro-Poulsen, H., Cellier, P., Cape, J. N., Horváth, L., Loreto, F., Niinemets, Ü., Palmer, P. I., Rinne, J., Misztal, P., Nemitz, E., Nilsson, D., Pryor, S., Gallagher, M. W., Vesala, T., Skiba, U., Brüggemann, N., Zechmeister-Boltenstern, S., Williams, J., O'Dowd, C., Facchini, M. C., de Leeuw, G., Flossman, A., Chaumerliac, N. and Erisman, J. W.: Atmospheric composition change: Ecosystems-Atmosphere interactions, Atmos. Environ., 43(33), 5193–5267, doi:10.1016/j.atmosenv.2009.07.068, 2009.

Klose, M. and Shao, Y.: Stochastic parameterization of dust emission and application to convective atmospheric conditions, Atmos. Chem. Phys., 12(16), 7309–7320, doi:10.5194/acp-12-7309-2012, 2012.

Klose, M. and Shao, Y.: Large-eddy simulation of turbulent dust emission, Aeolian Res., 8, 49–58, doi:10.1016/j.aeolia.2012.10.010, 2013.

Klose, M., Shao, Y., Li, X., Zhang, H., Ishizuka, M., Mikami, M. and Leys, J. F.: Further development of a parameterization for convective turbulent dust emission and evaluation based on field observations, J. Geophys. Res., 119(17), 10441–10457, doi:10.1002/2014JD021688, 2014.

Malcolm, L. P. and Raupach, M. R.: Measurements in an air settling tube of the terminal velocity distribution of soil material, J. Geophys. Res., 96, 15,275-15,286, 1991.

Petroff, A. and Zhang, L.: Development and validation of a size-resolved particle dry deposition scheme for application in aerosol transport models, Geosci. Model Dev., doi:10.5194/gmd-3-753-2010, 2010.

Seinfeld, J. H. and Pandis, S. N.: Atmospheric Chemistry and Physics: From Air Pollution to Climate Change, Wiley. [online] Available from: https://books.google.de/books?id=tZEpAQAAMAAJ, 2006.

Shao, Y.: Simplification of a dust emission scheme and comparison with data, J. Geophys. Res. D Atmos., 109(10), 1–6, doi:10.1029/2003JD004372, 2004.

Slinn, W. G. N.: Predictions for particle deposition to vegetative canopies, Atmos. Environ., 16(7), 1785–1794, doi:10.1016/0004-6981(82)90271-2, 1982.

Walcek, C. J., Brost, R. A., Chang, J. S. and Wesely, M. L.: SO2, sulfate and HNO3 deposition velocities computed using regional landuse and meteorological data, Atmos. Environ., 20(5), 949–964, doi:10.1016/0004-6981(86)90279-9, 1986.

Zhang, J. and Shao, Y.: A new parameterization of particle dry deposition over rough surfaces, Atmos. Chem. Phys., 14(22), 12429–12440, doi:10.5194/acp-14-12429-2014, 2014.

Zhang, L., Gong, S., Padro, J. and Barrie, L.: A size-segregated particle dry deposition scheme for an atmospheric aerosol module, Atmos. Environ., doi:10.1016/S1352-2310(00)00326-5, 2001.